# Enhancing Time Reading and Recording Skills in First-Grade Children with Learning Difficulties Using the “Clock Motor Game”

**DOI:** 10.3390/children10111748

**Published:** 2023-10-27

**Authors:** Aymen Hawani, Liwa Masmoudi, Omar Trabelsi, Mohamed Abdelkader Souissi, Anis ben Chikha, Maher Mrayah, Nizar Souissi, Santo Marsigliante, Mateusz Rozmiarek, Antonella Muscella

**Affiliations:** 1Higher Institute of Sport and Physical Education (Ksar Saïd), University of Manouba, Manouba 2010, Tunisia; aymen.hawani@issep.uma.tn (A.H.); anis.benchikha@issep.uma.tn (A.b.C.); meher.mrayeh@issep.uma.tn (M.M.); 2Physical Activity, Sport and Health, Research Unit (UR18JS01), National Observatory of Sport, Tunis 1003, Tunisia; trabelsi.omar@issepsf.u-sfax.tn (O.T.); abdelkader.souissi@isseps.usf.tn (M.A.S.); nizar.souissi@issep.uma.tn (N.S.); 3High Institute of Sport and Physical Education, University of Sfax, Sfax 3029, Tunisia; liwa.masmoudi@isseps.usf.tn; 4Research Laboratory, Education, Motricity, Sport and Health (EM2S), LR15JS01, High Institute of Sport and Physical Education, University of Sfax, Sfax 3038, Tunisia; 5High Institute of Sport and Physical Education of Kef, University of Jendouba, El Kef 7100, Tunisia; 6The High Institute of Sport and Physical Education of Gafsa, University of Gafsa, Gafsa 2112, Tunisia; 7Research Unit ECOTIDI (UR16ES10), Virtual University, Tunis 1073, Tunisia; 8Department of Biological and Environmental Science and Technologies (DiSTeBA), University of Salento, 73100 Lecce, Italy; santo.marsigliante@unisalento.it; 9Department of Sports Tourism, Faculty of Physical Culture Sciences, Poznan University of Physical Education, 61-871 Poznan, Poland; rozmiarek@awf.poznan.pl

**Keywords:** motor games, skill, reading and recording of time, mathematical learning difficulties, pupils

## Abstract

This study aimed to explore the effect of the motor game, “Clock Motor Games”, on the improvement of “Reading and Recording of Time” (RRT) in children with Grade 1 mathematical learning difficulties (MLDs). A within-school cluster-randomized intervention study was conducted with 232 children (aged 6–7 years) with limited physical education experience (0.7 ± 0.3 years). The participants were divided into two groups: a control group, which received conventional teaching on time without any additional motor activities, and an experimental group, which incorporated the concept of time with the “Clock Motor Game”, for 3 weeks. The Clock-Reading Test was administered before the intervention (T_0_), immediately after each session (T_1_), and five weeks after the intervention (T_2_) in both groups. The results demonstrated that the experimental group exhibited significantly greater improvements in RRT performance compared to the control group (U = 4416.5; *p* < 0.001; r = 0.3; medium effect). Additionally, the experimental group was more likely to show progress and less likely to experience regression or stagnation compared to the control group (25% vs. 38.4%). The findings suggest that practicing “Clock Motor Games” can positively contribute to the RRT ability in children with Grade 1 MLD.

## 1. Introduction

Acquiring the ability to navigate time using conventional time systems, such as clocks, is a crucial objective for primary school children. In fact, the skill of “Reading and Recording Time” (RRT) holds great significance as it equips us with essential life skills to efficiently plan our daily activities, stay organized, and thrive in a society governed by time [1].

This subject has received little scientific attention, and therefore, little is known about the development of RRT skills in children with learning difficulties [2]. For instance, children with mathematical learning difficulties (MLDs) have demonstrated increased challenges in clock reading [2,3,4], and this difficulty can also be one of the first signals pointing to dyscalculia in early school-age children [5].

Faced with this critical situation, a frequently adopted strategy by primary school teachers is to switch to digital clocks for children with MLDs in class [4]. However, this strategy does not resolve all difficulties with RRT [6]. Therefore, since 1997, Boulton-Lewis [7] has shown that it is necessary to use new strategies that facilitate teaching time concepts for Grades 1 to 3, especially for children with MLDs.

However, several studies have reported that children with MLDs encounter significant difficulties with RRT and perform noticeably worse than their peers [3,8]. Indeed, RRT shares certain similarities with numerical knowledge, mathematical facts, and mathematical procedures, as well as semantic memory retrieval, as proposed in the theory of subtypes in MLD by Geary and Hoard [9]. Children with MLDs are consistently performing worse in RRT on both analog and digital clocks across various grades in primary education. Moreover, an MLD is characterized by three common deficits: a procedural deficit, a semantic memory deficit, and a spatial deficit [9,10,11]. Most authors agree on the procedural and semantic memory subtypes within MLDs [12,13,14]. However, the spatial deficits remain unclear [15,16].

For the deficit in semantic memory among children with MLDs in RRT, several studies indicate that this results from verbal memory or long-term memory dysfunction, characterized by errors in the retrieval of arithmetic facts [9,17,18]. The procedural deficit in children with MLDs may be attributed to executive dysfunction and the strategies used to solve mental arithmetic tasks [19,20,21] and simple arithmetic problems [18].

The contribution of spatial ability to children’s spatial deficits with MLDs remains unclear in RRT. Spatial skills differ from other primary abilities [22] and are cognitive skills that can be improved over time [23]. Enhancing these skills can positively impact students’ mathematical abilities and reduce the risk of learning difficulties in elementary school [24]. Spatial ability is highly correlated with the visualization factor as it relies on executive functioning and visuospatial storage [25]. Indeed, spatial visualization has been considered a very important factor for success in the science, technology, engineering, and mathematics domain [26]. Therefore, it is imperative that training in spatial skills should be included in the studies of primary school pupils [27].

Despite there being a notable correlation between mathematics performance and clock-reading abilities at every grade level in primary education, it is important to note that this correlation is only moderate [6]. Especially in Grades 1 and 2, when children are taught the landmarks on an analog clock (hour, half hour, and quarter past/to), the correlation with mathematics achievement appears to be rather weak. Nevertheless, the results of this study suggest that difficulties with the acquisition of basic clock-reading skills may be predictive of mathematics difficulties [28]. When working with children with MLDs, it can be beneficial to incorporate interventions that explicitly address spatial skills. These can include activities promoting spatial reasoning, visualization, and the mental manipulation of geometric objects. Educational interventions targeting spatial abilities alongside mathematical instruction have shown promise in supporting the mathematical development of children [29] as well as the skills of children with MLDs in relation to RRT.

In this orientation, we are just at the beginning to find out more and more about the contribution of the motor game, “Clock Motor Games”, which can impact children’s abilities to understand spatial relationships, recognize patterns, and manipulate numbers, which are important skills for RRT in children with MLDs.

For this reason, we hypothesized that Clock Motor Games could give positive results in time reading skills for students with Grade 1 MLDs.

Therefore, the aim of this study was to investigate the impact of practicing “Motor Clock Games”, adapted to meet the needs of children with MLDs, on improving time reading skills for Grade 1 pupils. Furthermore, we compared the effectiveness of this method with conventional classroom teaching based on learning time.

## 2. Materials and Methods

### 2.1. Participants

A total of 232 pupils aged 6–7 years with a physical education experience of 0.7 ± 0.3 years were recruited to participate in the study. The participants included 113 females and 119 males from ten different Tunisian public schools, comprising 19 different school classes in the Tunis area (Table 1). The schools were selected based on similar demographic profiles, determined by the placement of the schools, and on grade-based graduation performance (Table 2).

All included participants were identified as having difficulties in mathematics (MLD) according to the results of three tests carried out in class after three weeks. Indeed, we chose pupils who had scores of less than 10/20 in the three tests.

Before the experiment, we obtained signed written consent forms from the parents of the students in which they clearly gave their permission for their children to participate in the present study. In addition, the present study was conducted according to the Declaration of Helsinki. The protocol was approved by the local Research Ethics Committee (CPP: N° 0105/2022; 17 November 2022).

An a priori power analysis was used to estimate the sample size (G*Power Version 3.1.9.4., Düsseldorf, Germany) based on the *t* test family (means: difference between two dependent means). The analysis output showed that a sample size of 228 subjects would be sufficient to identify significant differences (effect size = 0.24, power (1–β) = 0.95 with an actual power of 95.03 in this study.

### 2.2. Procedure

Figure 1 illustrates the experimental procedure. Prior to commencing the experiment, all participants received a comprehensive introduction to the overall setting, the equipment, and the specific experimental protocols. This measure was taken to minimize any potential learning bias throughout the study. During this experiment, the participants were tested at multiple time points: before the intervention (T_0_), immediately after each session (Ti_1_, Ti_2_, Ti_3_, Ti_4_, Ti_5_, and Ti_6_), and after five weeks of the intervention (T_2_). All testing sessions were scheduled in the morning, specifically between 10:00 a.m. and 12:00 p.m. Both groups followed the same teaching program but used different methods. The test sessions focused on RRT evaluation.

Regarding the teaching of the CMGG, a clock in the form of a circle was drawn on the ground, as shown in Figure 2. This circle has a diameter of 30 m and contains 12 radii. Each radius originates from the center of the circle and points towards each number indicated on the circle. All of the drawn lines have a width of 15 cm.

At the beginning of the game, children are positioned outside of the clock. Upon a signal, they enter the “hallway” of the clock, running in the direction indicated by the educator, either clockwise or counterclockwise. During the game, when the educator announces an oral or visual hour, the children enter the clock and attempt to align themselves on the corresponding lines, physically symbolizing the announced hour. The game concludes when the group successfully organizes and accurately reproduces the announced time in the quickest manner, determining the winning team.

The students who were included in the CON group, in a regular classroom, were seated at their desks, and the teacher conducted their lesson using various analog clocks displaying multiple times. These clocks have a diameter of 30 cm, and each time, the students must try to find the solution by indicating the exact time.

### “Clock-Reading Test” Procedure [2]

Mathematics is considered a foundational subject because arithmetic and logical reasoning form the basis for children’s acquisition of the ability to read a clock [2]. Indeed, the early foundations of mathematics for children are numerical and spatial representations [30]. Furthermore, the early foundations of mathematics can be viewed in terms of (a) primary preverbal number knowledge [31] and (b) secondary verbal or symbolic numbers [32].

In this orientation, we created a test based on the following:
The primary knowledge of preverbal numbers:
A system of objects capable of precisely representing small numbers (maximum of 3);An analog system capable of approximately representing larger sets.The secondary knowledge of symbolic numbers:
Verbal subitizing (i.e., the ability to map numerical words within small sets);Counting (i.e., the ability to recite a sequence of counting to 10 and to embrace the principles of 1-to-1 correspondence, the stable order, and the cardinality for the purpose of enumerating a set of objects);Knowing how to make comparisons of numerical magnitude (for example, understanding that three is less than four or that six is greater than five);Being able to perform the linear representation of numbers (i.e., understanding that numerical quantities increase linearly);Being able to carry out arithmetic operations (i.e., knowing how to transform small sets through subtraction and addition in both verbal and non-verbal contexts).

After three tests, children who do not meet the mean (10/20) are classified as having MLDs.

At the end of each session, the teacher gives each participant a paper containing three analog clocks displaying three different times (i.e., 10:15, 08:30, and 4:00; Figure 3). In order to objectively measure the children’s abilities to read the clock, we developed parallel and partially different clock-reading tests, taking into account the school levels of the pupils. This test showed an acceptable Cronbach’s alpha value of 0.74 (MLD: α = 0.70, NA: α = 0.73) [6]. Each student had 2 min to write their responses. After collecting the papers, their teachers proceeded to assess the students’ skills in RRT (Rapid Response Time) by awarding 1 point for a correct answer and 0 points for an incorrect answer. Hence, each participant received a score ranging from 0 to 3 at the end of each training session.

### 2.3. Statistical Analysis

A statistical analysis was performed using the Statistica 10 software (StatSoft, Cracow, Poland). This study investigated the impact of the Clock Motor Game on children’s clock-reading test performances, comparing the experimental group that was exposed to the CMG intervention with the control group. Within each group, Wilcoxon tests were conducted to assess the changes in performance from the pre-test (T_0_) to the post-test (T_2_). Additionally, a Mann–Whitney test was employed to compare the delta changes between the control and experimental groups. The effect size for non-parametric tests was calculated using the coefficient r (r = Z/√N) of Rosenthal [33] and was interpreted as follows: small (0.10–<0.30), medium (0.30–<0.50), and large (≥0.50) [34]. Furthermore, a 2 × 2 crosstabs procedure (chi-square test) was used to examine the association between group assignment and delta changes in performance from T_0_ to T_2_. The strength of the association between the two categorical variables was calculated using the odds ratio. The statistical analysis aimed to determine if the CMG intervention had a significant impact on the participants’ clock-reading abilities. Statistical significance was accepted at *p* < 0.05.

## 3. Results

Within the CON group, the Wilcoxon test revealed a significant improvement in the clock-reading test performance from T_0_ (2.5 ± 0.6 errors) to T_2_ (1.7 ± 0.9 errors; Z = 6.37, *p* < 0.001, r = 0.6, large effect). Thus, the CON group enhanced their time reading and recording skills over the course of the study using the conventional teaching approach. The children in the CMGG had a more significant improvement in performance from T_0_ (2.5 ± 0.7 errors) to T_2_ (0.9 ± 1 errors; Z = 8.21, *p* < 0.001, r = 0.75, large effect), experiencing a notable progress in their abilities. These results are illustrated in Figure 4.

The CMGG obtained significantly greater improvements in their performance compared to the CON group, as evidenced by the delta changes, i.e., the differences between the pre-test and post-test scores (0.85, 34% and 1.62, 64% for the CON group and CMGG, respectively; Mann–Whitney test: U = 4416.5; *p* < 0.001; r = 0.3; moderate effect) (Figure 5). Thus, the Clock Motor Game had a more substantial positive impact on the experimental group’s performance.

The association between group assignment (1 = CON group; 2 = CMGG) and delta changes in performance from T_0_ to T_2_ (1 = progress; 2 = stagnation or regression) was analyzed using the chi-squared test. A significant association was found (χ^2^ (1) = 4.82; *p* = 0.028). The odds ratio = 1.87, indicating that the children in the CMGG were 1.87 times more likely to have achieved progress than the children in the CON group. Thus, the findings highlight the potential benefits of the CMG in promoting positive outcomes in the clock-reading test (Figure 6).

## 4. Discussion

This study aimed to investigate how a “clock-motor game”, which was adapted to meet the needs of children with MLDs, improves first-grade children’s time-reading abilities. We also aimed to compare the effectiveness of this method with conventional classroom-based teaching of time reading.

The main findings revealed that (i) the CMGG achieved significantly greater improvements in their performances compared to the control group in the RRT (Reading and Recording Time) task and (ii) the participants in the experimental group (75% vs. 61.6%) were more likely to have achieved progress and less likely to have experienced regression or stagnation compared to the CON group (25% vs. 38.4%). Specifically, the CMG had a significantly positive impact on RRT in children with mathematical learning difficulties in Grade 1. Moreover, the CMGG committed fewer errors than the CON group during the learning sessions and after the end of the experiment.

First of all, it is argued that the difficulties with RRT (Reading and Recording Time) result from a combination of procedural and semantic memory deficits; children with mathematical learning difficulties struggle due to the need for a combination of retrieval and procedural strategies to accurately read complex clock times [9]. As a result, Boulton-Lewis [7] emphasizes the necessity of employing new strategies to facilitate teaching time concepts, especially for children with MLDs in Grades 1 to 3. Among these strategies, school-related physical activity interventions have been shown to reduce anxiety, increase resilience, improve well-being, and promote positive mental health in children [35]. These interventions may offer promising benefits for children with MLDs and help to enhance their understanding of time concepts. After the CMG session, the observed beneficial effects of the CMG on RRT may be attributed to physiological adaptations (e.g., hormonal regulation) induced by physical exercise [36]. In fact, the CMG can potentially contribute to enhancing children’s mental health, like the benefits of physical exercise. Previous research has demonstrated that low-to-moderate-intensity exercise, such as the CMG, can have positive neurological effects, leading to cognitive improvements and specifically stimulating mechanisms that enhance cognitive outcomes [37].

On the other hand, the presence of persistent primitive reflexes in school-aged children, indicating an immature neuromotor status, has been widely reported [38,39,40,41,42,43,44,45]. This condition may impair cognitive skills’ performance and could be a contributing factor to specific learning difficulties, such as RRT. In this context, the practice of the CMG has shown potential in addressing issues related to the retention of primitive reflexes in children and in enhancing the time reading and recording skills of children with MLDs [46].

Five weeks after the intervention (T_2_), the participants who were exposed to the CMG over the course of the study experienced notable progress in their abilities in RRT. Indeed, the errors number committed by the CMGG during the realization of the “Clock-reading test” [2] underwent a significant decrease compared to the CON group. This could be explained by the rapid recognition of important landmarks on the clock face drawn on the school playground during the CMG sessions, since children with MLDs have difficulties in retrieving mental representations from long-term memory [9,19], they cannot rely on retrieval strategies, and they do not gain any automation in RRT. More specifically, children with MLDs have problems usually related to retrieving memory representations [3]. With a carefully designed virtual environment, spatial skills and retrieving memory representations can be improved [47]. This means that with the CMG, we can act on the retrieving memory representations of children with MLDs in RRT tasks.

Another possible reason for this improvement is the spatial representation of time, which is partially mediated by the motor system and grounded in spatially directed movement [48], such as reading or writing. Spatial ability is highly correlated with the visualization factor as it relies on executive functioning and visuospatial storage [28]. These confirmations align with the characteristics of the CMG, which enable it to act on visuospatial storage and mediate the notion of RRT in children with mathematical learning difficulties.

The beneficial effects of the CMG on RRT highlighted in this study may be explained by the engagement of visuospatial working memory (VSWM) during the CMG practice by first-grade children with MLDs. Several studies have shown a positive correlation between VSWM and reading accuracy [49,50], reading comprehension [51], as well as performances in science, English, and especially mathematics [52,53]. Moreover, deficits in VSWM can affect children’s social interaction processes and even lead to social dysfunction [54], negatively impacting children’s mental health [55,56,57,58,59]. Therefore, improving VSWM ability is crucial for all children, especially for those with MLDs.

Improvements in VSWM through physical activity interventions have been confirmed by many studies [11,60,61,62,63,64], especially in children aged 4–11 years [65], such as the children in our population. For children aged 4–11 years, positive effects have been found for physical activity interventions characterized by (a) low to moderate exercise intensity; (b) exercise for over 30 min per session; (c) ideal types of exercise including aerobic exercises (i.e., jogging, brisk walking, and aerobics) and coordination exercises [65]; (d) the use of motor-cognitive dual-task training, which can increase the level of cognitive engagement [66]. All of these characteristics are well respected during the practice of the CMG for children on the playground.

Lastly, this study suggests that the CMG is a new strategy that can facilitate teaching time concepts to children with MLDs in Grade 1. The findings of the present study corroborate those of previous research conducted by Boulton-Lewis in 1997 [7], which suggested that it is necessary to use new strategies that facilitate teaching time concepts for school-age children in Grades 1 to 3. That said, the practice of the CMG allows for an increase in cognitive outcomes in children with MLDs.

The current investigation possesses certain constraints that warrant consideration during the interpretation of the results. To commence, this study exclusively encompassed first-grade children diagnosed with MLDs, thereby circumscribing the extent to which the findings can be extrapolated to encompass a broader spectrum of school-age children. Henceforth, forthcoming inquiries ought to contemplate enrolling participants across diverse scholastic tiers. Secondly, this study is predicated upon the implementation of the CMG as a form of physical activity and its associated cognitive contributions. Therefore, forthcoming research endeavors should contemplate the inclusion of physiological metrics (e.g., heart rate, oxygen consumption, indices of fatigue, etc.) in conjunction with additional cognitive assessments. Lastly, the present study refrained from conducting a gender-based comparative analysis among the children. Subsequent investigations should undertake an exploration of these variables, as they could conceivably exert influences on the amelioration of temporal comprehension and record-keeping proficiencies among school-age children, particularly those grappling with MLDs.

Despite these limitations, this study sheds light on the potential benefits of integrating motor activities into educational practices, emphasizing the importance of considering individual learning needs to foster essential life skills in children with learning difficulties. Furthermore, this study opens a promising way to study the pedagogical implications of practical lessons that combine different subjects (physics, science, mathematics, etc.) with physical education, especially for children with learning difficulties.

## 5. Conclusions

The findings of this study indicate that a duration of 6 weeks dedicated to engaging in “Clock Motor Games” activities within the school playground manifests a favorable influence on the aptitude for “Reading and Recording Time” among Grade 1 children afflicted with MLDs. Across all children, participation in motor games for learning purposes yielded a heightened proficiency in the domain of “Reading and Recording Time” compared to conventional lessons about learning about time in class.

Taking a pragmatic stance, educators within primary schools should contemplate the incorporation of motor games into educational endeavors aligned with the scholastic syllabus. This approach holds considerable potential in terms of involving children with MLDs, particularly given the abundance of occasions for physical engagement throughout the school week. Such opportunities encompass intervals between classes, sporting activities, physical education sessions, and even the act of commuting to and from school [67].

## Figures and Tables

**Figure 1 children-10-01748-f001:**
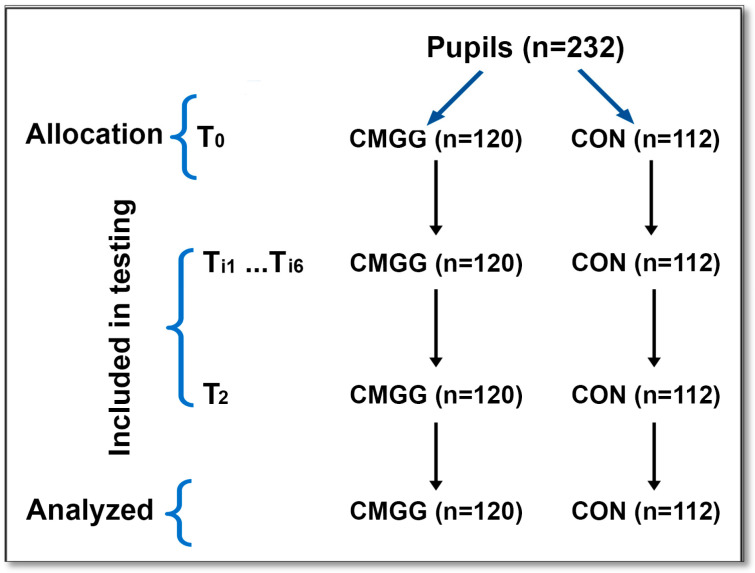
Flow diagram for the study. CMGG, Clock Motor Game group; CON, control group.

**Figure 2 children-10-01748-f002:**
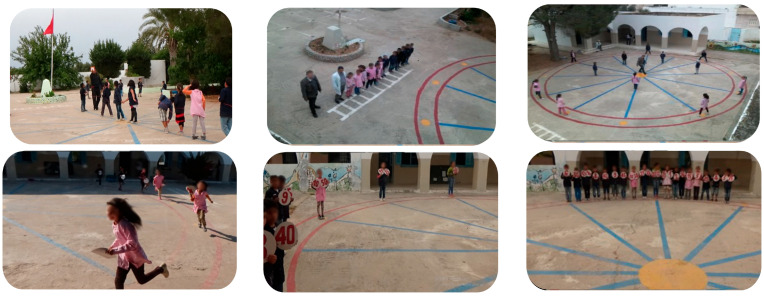
Clock Motor Game.

**Figure 3 children-10-01748-f003:**
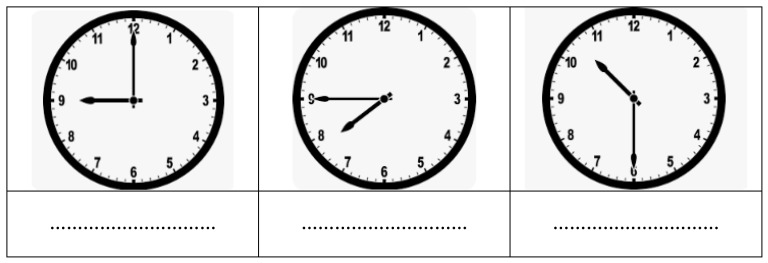
Pupils’ first-grade test.

**Figure 4 children-10-01748-f004:**
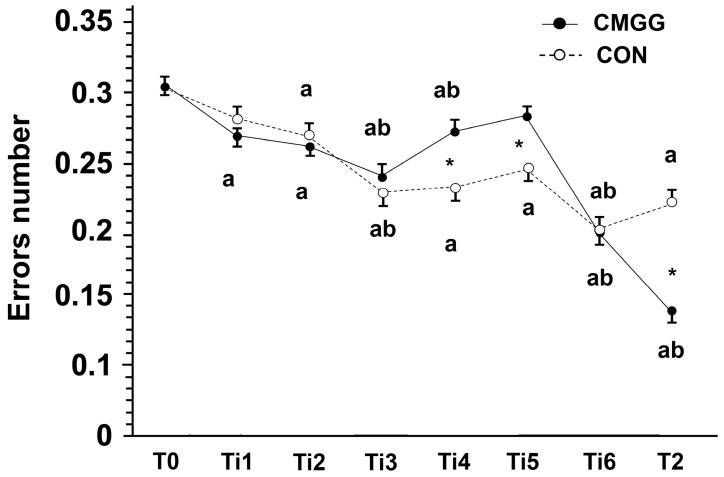
Variation of the error rate in the two groups. * Significant difference between groups; a means significantly different from T_0_; b means significantly different compared to the previous session.

**Figure 5 children-10-01748-f005:**
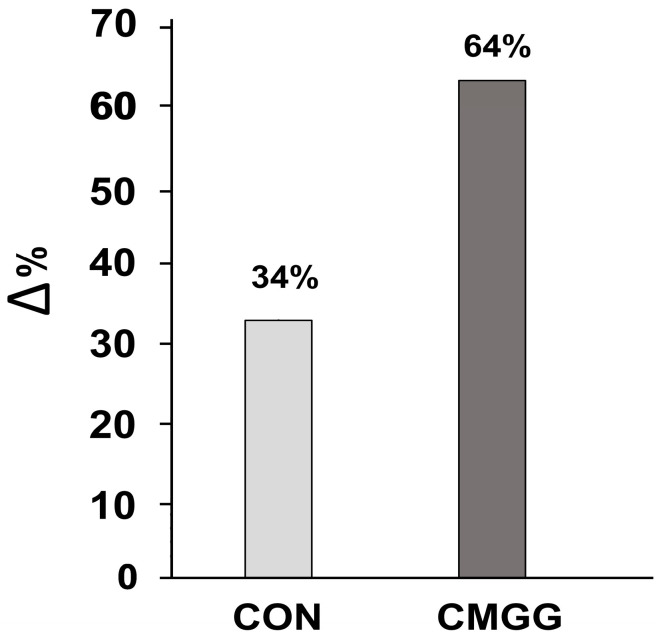
Clock-reading performance in the two groups.

**Figure 6 children-10-01748-f006:**
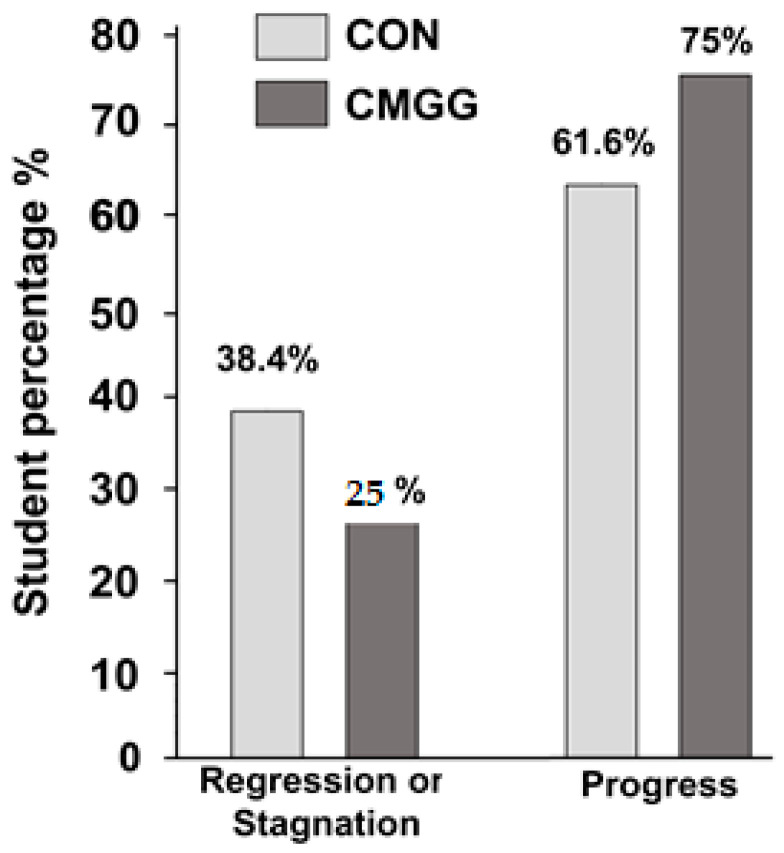
The association between group assignment and delta changes in performance from T_0_ to T_2_.

**Table 1 children-10-01748-t001:** Participants’ school affiliations.

	Public Primary Schools (Tunis Area)	School Code	Control Group	Clock Motor Game Group
**1**	Primary school “Al Madina”	100104	10	12
**2**	Primary school “Bab Bhar”	100304	13	11
**3**	Primary school “El Kabbaria”	100601	12	12
**4**	Primary school “Sidi El Bachir”	100702	10	13
**5**	Primary school “El Wardia”	100801	12	12
**6**	Primary school “2 Mars Lakania”	100817	13	11
**7**	Primary school “Hay Mohamed Ali Wardia”	100818	10	11
**8**	Primary school “Sidi Fathallah”	101408	12	13
**9**	Primary school “Nahej El ward”	101608	10	12
**10**	Primary school “Bab Souika”	100501	10	13

**Table 2 children-10-01748-t002:** Characteristics of the two groups.

	Control Group	Clock Motor Game Group
**Participants** (n)	112	120
**Age** (Years)	6.8 ± 0.5	6.6 ± 0.7
**Gender**	57 Male/55 Female	62 Male/58 Female
**Physical Education Experience**	0.7 ± 0.3 years	

## Data Availability

Data is contained within the article.

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
