# Peer review of "Enhancing Time Reading and Recording Skills in First-Grade Children with Learning Difficulties Using the “Clock Motor Game”"

_children, 2023, doi:10.3390/children10111748_

Round 1

Reviewer 1 Report

Comments and Suggestions for Authors

In chapter 2.3 statistical analysis please add that Spearman rank test was used

Line 173- please cite the reference where it proves that Spearman correlation of 0.6 is one with the large effect

Line- please cite the reference where it proves that Spearman correlation of 0.3 is medium effect

Author Response

Dear reviewer
We appreciate the time and effort that you dedicated to providing feedback on our manuscript and we feel grateful for the insightful comments and valuable suggestions. We have incorporated most of the suggestions made and carefully considered your concerns.

All changes are highlighted throughout the revised manuscript.

Point 1

In chapter 2.3 statistical analysis please add that the Spearman rank test was used.

Line 173- please cite the reference where it proves that Spearman correlation of 0.6 is one with a large effect

Response 1:

Spearman's rank test is not used in this study. However, the effect size for non-parametric tests was calculated by the coefficient r  of Rosenthal (1991) and was interpreted as follows:

small (0.10–<0.30), medium (0.30–<0.50), and large (≥0.50), (King et al., 2011).

Chapter 2.3. (statistical analysis) is replaced by the following paragraph:

“The statistical analysis was performed using the Statistica 10 software (StatSoft, Cracow, Poland). The study investigated the impact of the Clock Motor Game on clock-reading test performance, comparing the experimental group exposed to the CMGG intervention with the control group. Within each group, Wilcoxon tests were conducted to assess the changes in performance from the pretest (T0) to the posttest (T2). Additionally, a Mann-Whitney test was employed to compare the delta changes between the control and experimental groups. The effect size for non-parametric tests was calculated by the coefficient r (r=Z/√N) of Rosenthal (1991) and was interpreted as follows: small (0.10–<0.30), medium (0.30–<0.50), and large (≥0.50), (King et al., 2011). Furthermore, a 2 × 2 crosstabs procedure (chi-square test) was used to examine the association between group assignment and delta changes in performance from T0 to T2. The strength of the association between the two categorical variables was calculated using the odds ratio. The statistical analysis aimed to determine if the CMGG intervention had a significant impact on the participants’ clock-reading abilities. Statistical significance was accepted at p<0.05.”

Point 2

Line 183- please cite the reference where it proves that Spearman correlation of 0.3 is medium effect

Response 2:

Two bibliographic references have been added to the reference section:

Rosenthal, R. (1991). Meta-analytic procedures for social research (Rev. ed). Newbury Park: Sage Publications.

King, B. M., Rosopa, P. J., & Minium, E. W. (2011). Statistical reasoning in the behavioral sciences (6th ed.). John Wiley.

In the revised paper, we have done our best to improve the quality of the different sections of the manuscript, as requested by the reviewer. We hope that the changes made throughout the manuscript (highlighted in yellow) will meet your expectations.

Reviewer 2 Report

Comments and Suggestions for Authors

This study investigated the potential benefits of using mathematical instruction and motor games, specifically "Clock Motor Games," to support the mathematical development and skills of children with learning difficulties. In general, this study is well-developed with solid research methodology. However, I have a few comments which the authors need to address further. 

First, it would be appreciated to add a section of the literature review that includes prior studies relevant to this study. I know the authors mentioned these studies in the introduction. It would increase the readability by adding an additional section. Moreover, the introduction needs to have straightforward research questions and hypotheses. The current version does not have clear research questions and hypotheses for this study. 

Second, please justify the inclusion of students with difficulties in mathematics. How did you justify that students with less than 10/20 in the three tests are MLD? Is there any reference for the inclusion criteria? 

Third, since the findings support that the Clock Motor Games are helpful in improving students' performance, what are the pedagogical implications for further math education? It seems that it needs to combine different subjects (i.e., math and PE) to make it happen. Maybe the authors can address some pedagogical implications for practical teaching. 

Comments on the Quality of English Language

The quality of English is good but needs minor revisions. 

Author Response

Dear reviewer
We appreciate the time and effort that you dedicated to providing feedback on our manuscript and we feel grateful for the insightful comments and valuable suggestions. We have incorporated most of the suggestions made and carefully considered your concerns.

All changes are highlighted throughout the revised manuscript.

Point 1: First, it would be appreciated to add a section of the literature review that includes prior studies relevant to this study. I know the authors mentioned these studies in the introduction. It would increase the readability by adding an additional section. Moreover, the introduction needs to have straightforward research questions and hypotheses. The current version does not have clear research questions and hypotheses for this study.

Response 1:

Thanks for your comments. Done (line 102).

Point 2: Second, please justify the inclusion of students with difficulties in mathematics. How did you justify that students with less than 10/20 in the three tests are MLD? Is there any reference for the inclusion criteria?

Response 2:

In materials and methods (2.2.1. “Clock-reading test” procedure) we have justified the inclusion of students with difficulties in mathematics and added some references:

“Mathematics is considered a foundational subject because arithmetic and logical reasoning form the basis for children's acquisition of clock reading skills [2]. Indeed, the early foundations of mathematics for children are numerical and spatial representations [30]. Furthermore, the early foundations of mathematics can be viewed in terms of (a) primary preverbal number knowledge [31] and (b) secondary verbal or symbolic numbers [32].

In this orientation, we created a test based on:

  1. Primary Preverbal Number Knowledge
  • Object file system for precise representation of small numbers (3 or less)
  • Analogue magnitude system for approximate representation of larger sets
  1. Secondary Symbolic Number Knowledge
  • Verbal subitizing (mapping number words onto small sets)
  • Counting (reciting the count sequence to 10 and grasping principles of 1–1 correspondence, stable order, and cardinality to enumerate sets of objects)
  • Numerical magnitude comparisons (e.g., knowing that two is smaller than five or that five is larger than four)
  • Linear representations of numbers (understanding that numerical magnitudes increase linearly)
  • Arithmetic operations (transforming small sets through adding and subtracting in nonverbal and verbal contexts)

After three tests, children who do not meet the mean (10/20) are classified as having MLD”.

Point 3: Third, since the findings support that the Clock Motor Games are helpful in improving students' performance, what are the pedagogical implications for further math education? It seems that it needs to combine different subjects (i.e., math and PE) to make it happen. Maybe the authors can address some pedagogical implications for practical teaching.

Response 3:

At the end of Discussion, we added:

“Despite these limitations, this study sheds light on the potential benefits of integrating motor activities into educational practices, emphasizing the importance of considering individual learning needs to foster essential life skills in children with learning difficulties. Furthermore, this study opens a promising way to study the pedagogical implications of practical lessons that combine different subjects (physics, science, mathematics...) with physical education, especially for children with learning difficulties”.

We have done our best to improve the quality of the different sections of the manuscript, as requested by reviewer 2 and the manuscript has been proofread for syntax and grammatical errors. We hope that the changes made throughout the manuscript (highlighted in green) will meet your expectations.

Reviewer 3 Report

Comments and Suggestions for Authors

Dear Authors,

Interesting study from the perspective of the practice of physical activity and its interaction with executive mental skills in early schoolchildren. As the authors point out, other physiological variables could be evaluated, for example, and the results compared between both genders. Also, it is important to project the study towards other learning contents such as natural sciences, languages, etc.

Some considerations.

1. Maximum of 200 words, the summary must be shortened.

2. Check and correct keywords at: https://meshb.nlm.nih.gov/

3. The objective of the study should be made clearer; the purpose should be specifically stated at the end of the discussion section.

4. The type of test that was used must be described: its items, its scoring, categories, etc. (line 108).

5. Did the children give their consent to participate? Through a written or gestural symbol, they signed some type of document. This should be clarified (line 110).

6. How many participants belonged to each school? since table 1 does not clarify this doubt (line 117).

7. What test were applied, the questionnaires on mathematical tests? This aspect must be clarified. ...or was it the Burny test (ref. 6) and the CON group's small analog clock test? (line 127).

8. I understand that this figure shows the results of the Burny test (ref. 6) which was applied after each session for the CMGG group, and for the smaller analog clock test (30 cm diameter) in the CON group. Please clarify this question (figure 3).

9. Figure 4 repeats the same result that is in the text, I recommend removing it.

10. When using the Chi-squared test, it is most appropriate to talk about prevalence and not probabilities, since it has not been calculated using the OR or HR, for example. Therefore, it can be said that there was a prevalence of one group over another, but not greater probabilities of one group over the other. What do the authors think? I await your comments (line 193).

11. This objective must be made explicit at the end of the introduction section (line 203-205).

12. This is a projection of the study, rather than a limitation (line 285 - 286).

13. Some research strengths need to be added, for example, the Burny test (ref. 6). Indicate strengths at the end of the discussion section.

Thanks

Author Response

Dear reviewer
We appreciate the time and effort that you dedicated to providing feedback on our manuscript and we feel grateful for the insightful comments and valuable suggestions. We have incorporated most of the suggestions made and carefully considered your concerns.

All changes are highlighted throughout the revised manuscript.

Point 1: Maximum of 200 words, the summary must be shortened.

Response 1: Thank you. The abstract has been shortened.

Point 2: Check and correct keywords at: https://meshb.nlm.nih.gov/

Response 2: We have replaced the keywords Thanks.

Point 3: The objective of the study should be made clearer; the purpose should be specifically stated at the end of the discussion section.

Response 3: According to Reviewer's suggestion, we added the:

“Despite these limitations, this study sheds light on the potential benefits of integrating motor activities into educational practices, emphasizing the importance of considering individual learning needs to foster essential life skills in children with learning difficulties. Furthermore, this study opens a promising way to study the pedagogical implications of practical lessons that combine different subjects (physics, science, mathematics...) with physical education, especially for children with learning difficulties.”

Point 4: The type of test that was used must be described: its items, its scoring, categories, etc. (line 108).

We explained in detail, adding:

“Mathematics is considered a foundational subject because arithmetic and logical reasoning form the basis for children's acquisition of clock reading skills [2]. Indeed, the early foundations of mathematics for children are numerical and spatial representations [30]. Furthermore, the early foundations of mathematics can be viewed in terms of (a) primary preverbal number knowledge [31] and (b) secondary verbal or symbolic numbers [32].

In this orientation, we created a test based on:

  1. Primary Preverbal Number Knowledge
  • Object file system for precise representation of small numbers (3 or less)
  • Analogue magnitude system for approximate representation of larger sets
  1. Secondary Symbolic Number Knowledge
  • Verbal subitizing (mapping number words onto small sets)
  • Counting (reciting the count sequence to 10 and grasping principles of 1–1 correspondence, stable order, and cardinality to enumerate sets of objects)
  • Numerical magnitude comparisons (e.g., knowing that two is smaller than five or that five is larger than four)
  • Linear representations of numbers (understanding that numerical magnitudes increase linearly)
  • Arithmetic operations (transforming small sets through adding and subtracting in nonverbal and verbal contexts).

After three tests, children who do not meet the mean (10/20) are classified as having MLD”.

Point 5: Did the children give their consent to participate? Through a written or gestural symbol, they signed some type of document. This should be clarified (line 110).

Response 5: Children cannot sign consent to participate in this study due to their age (6-7 years old); then, after discussing the work protocol with their children, the parents signed the consent.

Point 6: How many participants belonged to each school? since Table 1 does not clarify this doubt (line 117).

Response 6: Participants were 178 females and 154 males from 10 different Tunisian public schools, comprising 19 different classes in the Tunis area. In Table 1, we have detailed the affiliations of the participating children, both in the control group and in the experimental group.

 Point 7: What test were applied, the questionnaires on mathematical tests? This aspect must be clarified. ...or was it the Burny test (ref. 6) and the CON group's small analog clock test? (line 127).

Response 7:

  • A) For mathematical tests:

We explained in detail, adding:

“Mathematics is considered a foundational subject because arithmetic and logical reasoning form the basis for children's acquisition of clock reading skills [2]. Indeed, the early foundations of mathematics for children are numerical and spatial representations [30]. Furthermore, the early foundations of mathematics can be viewed in terms of (a) primary preverbal number knowledge [31] and (b) secondary verbal or symbolic numbers [32].

In this orientation, we created a test based on:

  1. Primary Preverbal Number Knowledge
  • Object file system for precise representation of small numbers (3 or less)
  • Analogue magnitude system for approximate representation of larger sets
  1. Secondary Symbolic Number Knowledge
  • Verbal subitizing (mapping number words onto small sets)
  • Counting (reciting the count sequence to 10 and grasping principles of 1–1 correspondence, stable order, and cardinality to enumerate sets of objects)
  • Numerical magnitude comparisons (e.g., knowing that two is smaller than five or that five is larger than four)
  • Linear representations of numbers (understanding that numerical magnitudes increase linearly)
  • Arithmetic operations (transforming small sets through adding and subtracting in nonverbal and verbal contexts)

After three tests, children who do not meet the mean (10/20) are classified as having MLD”.

  1. B) For the Clock-reading test” (Burny, E. et al, 2013), we have added a figure 3 for additional explanation.

Point 8: I understand that this figure shows the results of the Burny test (ref. 6) which was applied after each session for the CMGG group and for the smaller analog clock test (30 cm diameter) in the CON group. Please clarify this question (figure 3).

All the pupils in both samples (CMGG/CON) completed a specific test of reading and recording Time (Figure 3).

To measure children’s clock-reading abilities, parallel and partly different clock-reading tests were developed, considering the grade level of the pupils involved. This test showed an acceptable Cronbach’s alpha value of 0.74 (MLD: α= 0.70, NA: α=0.73) [6]. Each student has 2 minutes to write their response. After collecting the papers, their teachers proceeded to assess the student’s skills in RRT (Rapid Response Time) by awarding 1 point for a correct answer and 0 points for an incorrect answer. Hence, each participant should receive a score ranging from 0 to 3 at the end of each training session.

Point 9: Figure 4 repeats the same result that is in the text, I recommend removing it.

Response 9: Thank you, but I think that Figure 4 allows us to quickly and effectively show the variation in the error rate in the two groups, making it easy for readers to understand.

We did our best to enhance the quality of the different sections of the manuscript, as requested by the Reviewer. We hope that the amendments made throughout the manuscript (highlighted in light blue) will meet your expectations.

Round 2

Reviewer 2 Report

Comments and Suggestions for Authors

I would like to thank the authors for their efforts to revise this manuscript. The authors fully addressed my comments in the revised manuscript, and I am satisfied with the authors' revisions and responses. I endorse the publication of this work on Children. Congrats 

Comments on the Quality of English Language

The authors fully addressed my comments in the revised manuscript, and I am satisfied with the authors' revisions and responses.

Author Response

I would like to thank the reviewer for his valuable comments. Thanks to his suggestions the revised manuscript was greatly improved.

Reviewer 3 Report

Comments and Suggestions for Authors

Dear authors,

Thanks for responding to the corrections, these are relatively good answers.

Regards. 

Author Response

(The authors gave the same response as above.)
